# Bio-Based Packaging Materials Containing Substances Derived from Coffee and Tea Plants

**DOI:** 10.3390/ma13245719

**Published:** 2020-12-15

**Authors:** Olga Olejnik, Anna Masek

**Affiliations:** Institute of Polymer and Dye Technology, Faculty of Chemistry, Lodz University of Technology, ul. Stefanowskiego 12/16, 90-924 Lodz, Poland; olejnik.olg@gmail.com

**Keywords:** polyphenol, polylactide, biodegradable, bio-based, UV indicator, packaging, starch, caffeic acid, green tea extract

## Abstract

The aim of the research was to obtain intelligent and eco-friendly packaging materials by incorporating innovative additives of plant origin. For this purpose, natural substances, including green tea extract (polyphenon 60) and caffeic acid, were added to two types of biodegradable thermoplastics (Ingeo™ Biopolymer PLA 4043D and Bioplast GS 2189). The main techniques used to assess the impact of phytocompounds on materials’ thermal properties were differential scanning calorimetry (DSC) and thermogravimetry (TGA), which confirmed the improved resistance to thermo-oxidation. Moreover, in order to assess the activity of applied antioxidants, the samples were aged using a UV aging chamber and a weathering device, then retested in terms of dynamic mechanical properties (DMA), colour changing, Vicat softening temperature, and chemical structure, as studied using FT-IR spectra analysis. The results revealed that different types of aging did not cause significant differences in thermo-mechanical properties and chemical structure of the samples with natural antioxidants but induced colour changing. The obtained results indicate that polylactide (PLA) and Bioplast GS 2189, the plasticizer free thermoplastic biomaterial containing polylactide and starch (referred to as sPLA in the present article), both with added caffeic acid and green tea extract, can be applied as smart and eco-friendly packaging materials. The composites reveal better thermo-oxidative stability with reference to pure materials and are able to change colour as a result of the oxidation process, especially after UV exposure, providing information about the degree of material degradation.

## 1. Introduction

Packaging is an inseparable element of every product, including food [1], cosmetics [2], electronics [3], and clothing [4]. The main function of packaging is protecting its content from the impact of the external factors and keep the goods safe for a specified period of time [5]. It can also act as a marketing tool which helps to communicate with consumers [6]. Another aspect in the traditional classification of the packaging function is its convenience, which means that packaging should provide ease of use and save consumers’ time [7]. The last function concerns containment, because products of different shape and size should preserve their dimensions [8]. Sometimes, one function results in another. For example, the communication role of the package plays a part in the enhancement of food protection and convenience [9]. Moreover, a lot of emphasis is placed not only on functionality, but also on sustainability [10,11], aesthetics [12], and innovative solutions [13]. For that reason, material selection for packaging application is of great importance.

In 2019 the global production of bioplastics, including bio-based as well as biodegradable materials, amounted to about 2.11 million tons, which represented only about 1% of the total thermoplastics market [14]. Nevertheless, consumers as well as producers have recently become more aware of the importance of the natural environment protection, and therefore the production of bioplastics is still growing. Starch and poly(lactic acid) (polylactide, PLA) are the most popular biodegradable and simultaneously bio-based polymers representing respectively 21.3% and 13.9% of the global production capacities of bioplastic. A significant part of these polymers is dedicated to packaging production [15]. Polylactide (PLA) is similar to PET due to its good transparency, brightness, printing vulnerability, and good surface adhesion, and has better mechanical properties, including stiffness, which makes this polymer a good material for rigid packaging with thin walls [16]. Moreover, this thermoplastic is definitely dedicated to food contact because of its safety [17]. Another advantages of PLA as a packaging material is its superior resistance to greasy substances. Furthermore, its barrier properties are limited, thus polylactide can be applied as a breathable material for vegetables, fruit and bread packing, where moisture and vapor permeability occur [18]. However, the unwaveringly high price of this material has limited its wider application [19]. In comparison to polylactide, starch is cheaper and its processing by traditional equipment is definitely easier [20]. Nevertheless, the thermoplastic form of this polymer cannot be utilized as a self-sufficient packaging material due to its weakness, lower mechanical properties, and high moisture sensitivity. For that reason, many starch-based biodegradable materials have already been developed. One of the most satisfying ideas was the creation of bio-based and biodegradable composites by polylactide and thermoplastic starch blending [21,22]. This concept enabled scientists to achieve a compromise between the suitable mechanical properties and the acceptable cost of the packaging material. The polylactide- and starch-based composites are now widely available commercially and one of their main representatives is a material called Bioplast [23]. The manufacturer declares that this composite consists of 69% bio-based carbon and the whole material is completely compostable, but the exact information about the composition is not provided [24]. Nevertheless, some research data can be found in literature, where it has been demonstrated that one of the Bioplast materials, namely Bioplast GS 2189 is composed of approximately 55% PLA and 45% of starch [25]. This kind of material stands a chance to arise significantly on the market and replace the conventional thermoplastics in the future, hence it is these composites that have been chosen as a polymeric matrix in the current research.

The eco-friendly characteristics of the material in combination with the innovative solutions can result in the creation of a new packaging generation [26]. Thus, nowadays, we can find more examples of active as well as intelligent packages [27,28]. The active materials contain components able to release/absorb the chemical compounds into or from the food/environment surrounding the food to extend the shelf life, maintain the condition or improve the quality of packed food. Different strategies for active foods are known, including oxygen removal, temperature, and moisture control. [29]. Unlike active packaging, the intelligent one can detect the conditions in a package and inform the users about them [30]. In such smart systems, examples of different sensors and indicators can be presented, including biosensors [31], gas sensors [27], chemical sensors [32], electronic nose [33], freshness indicators [34], time temperature indicators [35], integrity indicators, and radiofrequency identification [29]. The presented systems are not only helpful for keeping products fresh but also are attractive for potential consumers, who are usually interested in innovative solutions.

Taking into account the safety of the human beings and the natural environment, the selected extra compounds dedicated to packaging materials should also not be toxic or eco-unfriendly. There are many examples of natural compounds with antioxidant properties which can be applied in polymeric material as additional substances [36]. Many of them reveal also extra functions, including, a thermal or UV stabilization effect [37,38], coloring properties [39] or may act as aging indicators [40]. For example, the incorporation of natural flavonoids, such as quercetin, can prevent material from the harmful impact of external factors as well as may inform consumers about the product’s condition [41]. The food packaging sector especially is showing an increasing interest in the development of innovative solutions, including new processes and materials. Natural substances, including phenolic compounds from plant extracts stand a chance of playing a significant role in an active as well as intelligent food packaging. Nevertheless, most of proposed natural additives are derived from fruit or vegetables [42]. Here, we present the potential of polyphenolic extract derived from green tea, as well as a phenolic acid present in coffee plant, as multifunctional auxiliary substances dedicated to pro-ecological packaging materials. These substances reveal antioxidant properties [43,44], thus seem to be good active ingredients for polymers, especially biodegradable ones dedicated to food packaging industry. Moreover, catechins and caffeic acid present in tea and coffee extracts change their colour after UV exposure [45], thus these natural phenolic compounds after incorporation to the polymer could inform about the exposure of the packaging and its content to the sunlight. Therefore, a combination of polymeric material with selected substances of plant origin can result in the creation of intelligent food packaging.

## 2. Materials and Methods

### 2.1. Components

One of the main objects of the research was polylactide (Ingeo™ Biopolymer PLA 4043D)—a product from NatureWorks^®^ LLC (Minnetonka, MN, USA) in a pellet form with the density of 1.25 g/cm^3^ containing 4.8% D-lactide. The average molecular weight of the selected PLA amounts to 200 kDa. The glass transition temperature of this biopolymer is about T_g_ = 328.15–333.15 K and the melting point equals T_m_ = 418.15–433.15 K. The melt flow index (MFI) of the tested polylactide is 6 g/10 min. Another material chosen as a matrix for the tested composites was compostable commercial thermoplastic Bioplast GS 2189 provided by BIOTEC GmbH & Co. KG (Emmerich on the Rhine, Germany). In the technical data sheet of this polymer, it can be found the information that this material in a granulate form is easy flowing and is characterized by MFI of about 35 g/10 min (463 K, 2.16 kg) and density of 1.35 g/cm^3^. The producer declares the bio-based carbon share of the whole formulation reaches 69%. The precise composition is not revealed by the producer but the test results available in literature show that Bioplast consists of approximately 55% PLA and 45% of starch. The molecular weight of the material’s polylactide part is about 55 kDa. The glass transition temperature amounts to approximately 333–335 K and the melting point is below 473 K. The selected antioxidants of plant origin, including polyphenon 60 from green tea purchased from Sigma Aldrich (a part of the Merck group, Germany) extract containing a mixture of polyphenolic compounds containing at least 60% total catechins) and caffeic acid (from Sigma Aldrich, with purity of ≥98.0% (HPLC) and molecular weight of 180.16 g/mol) were used as additional substances for Ingeo Biopolymer 4043D and Bioplast GS 2189.

### 2.2. Composites Preparation

The samples were formed using extrusion in a laboratory extruder provided by Zamak Mercator (Skawina, Poland). The extruder equipped with a single screw characterized by a diameter of D = 25 mm (L/D = 24) works in a horizontal position. The pellet of commercial thermoplastics, including Ingeo™ Biopolymer 4043D and Bioplast GS 2189 was initially dried at the temperature of 343 K for 24 h and later combined in laboratory beakers with selected antioxidants in a powder form in the following mass ratio: 3 parts of selected antioxidant per 100 parts of polymer. The prepared mixtures of polymer granulate and natural additives were finally extruded with the following process conditions: the screw rotation speed of 40 rpm, the pressure of 17 atm, the temperature of 453 K in all sections (feed section, compression section and metering section). The final samples in the shape of long flat stripes were formed by a special cylinder head, then cooled in the air at room temperature and ultimately cut into small pieces (150 mm long, 25 mm wide and 1 mm thick stripes) for further tests. The obtained composites described in Table 1 were subsequently measured in terms of different physico-chemical properties, then aged using a UV aging device and a weathering chamber, and finally retested.

### 2.3. Differential Scanning Calorimetry (DSC)

Differential scanning calorimetry (DSC) measurements were carried out using a Mettler Toledo^®^ DSC1 instrument (TA 2920; TA Instruments, Greifensee, Switzerland). This device with a STARe System and Gas Controller GC10 was calibrated using n-octane and indium standards. The DSC measurements was essential for obtaining all information about the temperature ranges of the characteristic phase changes, enthalpy changes (ΔH), glass transition temperature (T_g_), cold crystallization temperature (T_cc_) and melting temperature (T_m_). Moreover, the oxidation temperature (T_o_) was crucial for determining the effectiveness of the applied natural antioxidants. Every tested material with the weight of approximately 8–9 mg was settled in an open melting pot made of aluminum and then heated from 273 K to 473 K at the heating speed of 20 K/min under an argon atmosphere to erase the thermal history. The samples were held under the temperature of 473.15 K for 10 min and next cooled to 273 K. After this step, the gas was turned from argon to air with the flow velocity of 50 mL/min and the composites were heated to 623 K.

### 2.4. Thermogravimetric Analysis (TGA)

The thermal degradation process of the tested materials was detected by means of Thermogravimetric Analysis (TGA) using a TGA/DSC1 apparatus provided by Mettler Toledo^®^ (TA Instruments, Greifensee, Switzerand) calibrated on the basis of indium and zinc standards. The degradation process concerns the mass loss of a sample in a function of rising temperature. The measurement was performed in a temperature range of 298–873 K, where specimens were heated with the speed of 288 K/ min under an argon atmosphere. The flow rate amounted to 45 mL/min. In this test, crucibles made of ceramic (polycrystal aluminium oxide) characterised by a volume of 70 μL were used.

### 2.5. Dynamic Mechnical Analysis (DMA)

The dynamic mechanical properties analysis was conducted using an ARES G2 rotational rheometer by TA Instruments (New Castle, DE, USA). The measurement was carried out with an increasing angular frequency from 0.1 to 628 rad/s at the temperature of T = 453 K and oscillation strain of 1%. Both storage (G’) and loss modulus (G”) were measured as a function of the angular frequency respectively.

### 2.6. Colour Measurements

The colour measurement of the prepared composites was performed before and after different aging tests in accordance with the PN-EN ISO 105-J01 Standard using UV-VIS CM-36001 spectrophotometer provided by Konica Minolta Sensing Inc. (Osaka, Japan). This apparatus measures the signal reflected from the sample’s surface and converts it into the impression of the colour perceived by the human eye. The results are presented in the CIE-Lab space, where the colour is expressed by three values as follows: L-represents the lightness from black (0) to white (100), a-coordinate representing tones from green (−) to red (+), b-parameter corresponding with tones from blue (−) to yellow (+). Based on the obtained results, the colour difference (dE), whiteness index (W_i_), chroma (C_ab_) and hue angle (h_ab_) values were determined using Equations (1)–(4). These parameters were indispensable for estimating the colour changing as a result of incorporating the selected natural antioxidant into pure biomaterials as well as in the effect of the tested samples’ aging. The values Δa, Δb, and ΔL utilized in these equations were calculated as the difference of a, b, and L parameters between the tested sample and the referential one.
(1)$$dE=\sqrt{{\left(∆a\right)}^{2}+{\left(∆b\right)}^{2}+{\left(∆L\right)}^{2}}$$
(2)$$W_{i}=100-\sqrt{a^{2}+b^{2}+{\left(100-L\right)}^{2}}$$
(3)$$C_{ab}=\sqrt{a^{2}+b^{2}}$$
(4)$$h_{ab}=\left\{\begin{array}{c} arctg\;\left(\frac{b}{a}\right),\;when\;a>0\cap b>0\\ 180{}^{\circ}+arctg\;\left(\frac{b}{a}\right),\;when\;\left(a<0\cap b>0\right)\cup (a<0\cap b<0)\\ 360{}^{\circ}+arctg\left(\frac{b}{a}\right),\;when\;a>0\cap b<0 \end{array}\right.$$

### 2.7. Vicat Softening Temperature

The Vicat softening temperature is a specific parameter for polymers, for instance polylactide, which contain a crystalline as well as an amorphous phase. According to the definition, this parameter is a temperature where a stainless steel needle, with a circular cross-section and a surface area of 1 mm^2^ immerses into the sample to a depth of 1 mm under a defined load during the heating process. In this research the Vicat softening temperature measurement was conducted based on the PN–EN ISO 306 and ASTM D 1525 standards using D–Vicat.HDT/3/400FA device provided by Peter Huber Kältemaschinenbau GmbH (Offenburg, Germany). The test was performed providing a heating rate of 393 K/h and a load of 10 N. The samples were prepared by cutting the material into 1-mm high small pieces with the square-shaped 10 mm long base side and placed in 3-part stacks.

### 2.8. Attenuated Total Reflectance Fourier Transform Infrared Spectroscopy (ATR-FTIR) Analysis

The Attenuated Total Reflectance Fourier Transform Infrared Spectroscopy (ATR-FTIR) analysis was carried out utilizing Nicolet 6700 FT-IR Spectrometer by Thermo Scientific (Waltham, MA, USA) with diamond Smart Orbit ATR Smart iTX sampling equipment. The FTIR spectra was obtained in the range of 4000–400 cm^−1^ using 64 scans at a resolution of 4 cm^−1^. Based on the received spectra, the material changes occurring as a result of aging can be detected, and thus the effectiveness of selected natural antioxidants can be estimated.

### 2.9. UV Exposure

The prepared samples were aged in a UV 2000 device (Atlas Material Testing Technology LLC, Mt Prospect, IL, USA). The UV aging test was divided into two alternately repeating cycles. The first part, called daily cycle, was characterized by radiation intensity of E = 0.7 W/m^2^, temperature of T = 333 K, and duration of 8 h. The specific conditions of the second one, called night cycle, were: UV radiation absence, temperature of T = 323 K and duration of 4 h. The complete UV aging time of the tested specimens amounted to 70 h. This period was sufficient for observing the changes that had occurred in the tested materials.

### 2.10. Weathering Aging

The weathering aging test of the prepared samples was carried out in accordance with the PN-EN 4892-2 standard entitled “Plastics-Methods of exposure to laboratory light sources-Part 2: Xenon arc lamps” The aging was conducted using a Weather-Ometer Ci 4000 chamber (Atlas Material Testing Technology LLC, Chicago, IL, USA) with a xenon lamp. The samples were placed in special frames made of stainless steel. The process was divided into two alternately repeating parts. The first part, named daytime cycle, was characterized by radiation intensity of E = 0.3 W/m^2^ over a radiation range of 300–400 nm, temperature of 333 K, humidity of 80% with rainwater on, and duration of 240 min. The second cycle was simulating nighttime and had the conditions as follows: no radiation, temperature of 323 K, humidity of 60%, rainwater off, and duration of 120 min. The complete UV aging time of the tested specimens amounted to 70 h. This amount of time was sufficient for observing the changes that occurred in the tested materials.

## 3. Results and Discussion

### 3.1. Differential Scanning Calorimetry (DSC) Curves Interpretation

The thermo-oxidative stabilization of the tested materials, including pure polylactide (PLA) and starch-containing polylactide (sPLA) was detected using the differential scanning calorimetry (DSC). The obtained information about characteristic phase changes, such as enthalpy changes (ΔH), glass transition temperature (T_g_), cold crystallization temperature (T_cc_), melting temperature (T_m_) and oxidation temperature (T_o_) were compared with measurement results of the same biomaterials enriched with a green tea extract (polyphenon 60) and caffeic acid. All the results obtained are presented in Figure 1 and Table 2.

Starch-containing polylactide (sPLA, Bioplast GS 2189) has already been researched by Santos [25] using differential scanning calorimetry, where only polylactide’s melting as well as crystallization peaks were visible and indicated that the starch present in the material had an amorphous structure. Therefore, only the results related to the PLA part could be analyzed. In the current research, the results obtained are also related to the PLA content of the composite. These values were analyzed and compared with pure amorphous type PLA (4043D) and with the same bioplastics enriched with natural antioxidants. The addition of selected natural antioxidants to PLA did not modify the parameters of characteristic phase changes significantly, including glass transition temperature (T_g_) or melting point (T_m_) but the incorporation of caffeic acid induced the crystallization process at a slightly lower temperature (404 K) than in the case of pure polylactide (409 K). On the other hand, adding the selected substances of plant origin to starch-containing polylactide (sPLA) brought about visible changes in the material’s melting point (T_m_) and a slight plasticization effect can be noticed. This occurrence was the most visible in the case of polylactide containing starch with caffeic acid, where the melting point changed from 432 K to 425 K after incorporating this substance into polymer matrix. A similar phenomenon of an increase in melting point was observed by Francesca Luzi et al. [46] after adding caffeic acid to poly(vinyl alcohol-co-ethylene) The plasticizing effect of natural phenolic acid in polymer matrix is not as strong as of typical widely used plasticizers but a proper modification of natural phenolic acids can result in obtaining desirable plasticizers dedicated to biocomposites. Such modifications were performed by Samir Kasmi et al. [47], where ferulic acid derivatives were applied as plasticizers dedicated for PLA. The most important results relate to oxidation process, where all materials enriched with natural antioxidants revealed higher oxidation temperature (T_o_), therefore the enhanced thermo-oxidative stabilization effect was visible. The addition of green tea extract to PLA matrix caused a significant increase in T_o_ from 500 K to 546 K, while the same portion of caffeic acid brought about a growth to 552 K. In the case of the starch - containing polylactide (sPLA), the improvement of this parameter was also noticeable, where T_o_ increased from 486 K to 548.7 K after green tea extract incorporation and to 556 K after adding the same amount of caffeic acid. Either caffeic acid or green tea extract containing a significant amount of catechins and other flavonoids reveal antioxidant activity, mainly by free radical scavenging, which was proven in many researches [43,48]. The DSC analysis also confirmed antioxidant properties of selected natural substances by demonstrating delayed oxidation process of tested materials at an increased temperature.

The delayed oxidation process detected by differential scanning calorimetry can be explained by mechanism of radical scavenging activity via hydrogen transfer [49], presented in Figure 2. This mechanism is characteristic of catechins and other polyphenols present in green tea extract applied in this research. The caffeic acid also is a natural phenolic compound with antioxidant properties, and thus reveals a similar stabilizing effect. 

### 3.2. Thermogravimetric Analysis (TGA) Results of Selected Biomaterials with Substances of Plant Origin

The thermal stability of the created materials was estimated using Thermogravimetric Analysis (TGA). The results of TGA are presented in Table 3 and Appendix A. The decomposition of PLA materials has one stage, while in case of starch-containing polylactide (sPLA) about three stages can be observed. The first stage between 526 K and 643 K is responsible for PLA degradation. The temperatures of this stage are lower than detected by Atlli [24] or Santos [25], which is probably caused by the extrusion process, where material was treated by high temperature. The second stage between 643 K and 699 K consists in the starch decomposing and degradation occurring above 699 K responded to possible mineral waste contained in the starch. The pure amorphous PLA starts to decompose at higher temperatures but degrades completely and faster in comparison to starch-containing polylactide (sPLA). Based on the data obtained, the thermal stabilization effect is observed clearly in case of starch-containing polylactide (sPLA) matrix after the incorporation of green tea extract but the same amount of this substance deteriorated the thermal stability of pure polylactide by a decrease in the temperatures at the beginning of the first stage of the material’s weight loss. The addition of caffeic acid did not change the thermal stabilization effect significantly in the case of polylactide, while starch-containing polylactide (sPLA) lost its weight faster in the presence of the same substance.

### 3.3. Dynamic Mechanical Properities of PLA and PLA Containing Starch (sPLA) with Green Tea Extract and Caffeic Acid before and after UV Aging and Weathering

The dynamic mechanical analysis was useful for estimating the impact of natural antioxidant addition on a selected polymeric matrix. The results are presented in Figure 3 and Figure 4. Moreover, the resistance of the tested biomaterials to different aging types, i.e., UV exposure and weathering was also investigated (Appendix A). The measurement was performed at the temperature of T = 453 K. The rheological behavior of PLA differs from the starch-containing polylactide (sPLA). Nevertheless, the loss modulus prevails in these materials, which is characteristic of viscous properties. In the case of PLA material (Figure 3a) the crossover of G’ and G” curves at the high angular frequency of about 199 rad/s can be noticed. This crossover is equivalent to a gel point of the material. A similar appearance was detected by Hernández-Alamilla [50]. In contrast to PLA, the Bioplast GS 2189 (starch-containing polylactide) predominantly shows a viscous behaviour (G” > G’) in the whole angular frequency region studied and such phenomenon was not found in starch-containing polylactide (sPLA) composites. (Figure 3d). The presence of starch, where significant interactions between starch and PLA appear, prevents from gel point occurrence, thus starch-containing polylactide (sPLA) material does not reveal properties of a solid state even at the high frequency. In Figure 4 the difference of complex viscosity between PLA and starch-containing polylactide (sPLA) materials is visible, where starch-containing polylactide (sPLA) materials are characterized by lower complex viscosity. This phenomenon is also related to the presence of starch and/or compatibilizer in composite’s matrix which acts as plasticizer and causes a decrease in this parameter. According to Appendix A, pure PLA is resistant to weathering but UV exposure caused significant changes in G’ and G” as well as brought about a decrease in complex viscosity (Appendix A). The crossover was moved to higher frequency (398 rad/s), which is related to lower entanglements among the molecules. The addition of green tea extract did not change the rheological behavior (Figure 3a) but enhanced PLA resistance to UV irradiation (Figure 3b). The rheological curves of PLA in the presence of green tea extract are almost the same after different types of aging in comparison to the unaged material (Figure 3). Caffeic acid added to PLA matrix caused a plasticization effect, which is visible in lower values of complex viscosity (Figure 3a). The G’ and G” curves of PLA with caffeic acid were also changed in comparison to pure PLA. These results correlate with thermal analysis, where a lower glass transition temperature as well as melting point of polylactide (PLA) and starch-containing polylactide (sPLA) after the addition of caffeic acid were observed. The crossover of these moduli was moved to a substantially higher frequency and could not be detected. The material enriched with caffeic acid after weathering and UV aging seems to maintain the rheological behavior but only at higher frequency values (Appendix A). Analysing the moduli in lower frequency, the modifications compared to unaged PLA with caffeic acid are visible (Figure 3a). The complex viscosity in the same frequency range also decreased (Figure 4a), therefore this antioxidant turned out to be a less effective PLA stabilizer than green tea extract.

Starch-containing polylactide (sPLA) exposed to the UV radiation similarly to PLA is not stable, which was revealed by the higher storage modulus (G’) and loss modulus (G”) (Appendix A) as well as increased complex viscosity (Appendix A) after UV aging. The incorporation of green tea extract to polylactide containing starch (sPLA) matrix changed dynamic mechanical properties without enhancing its UV stability (Figure 3d). The addition of caffeic acid to starch-containing polylactide caused a slight increase in G’ as well as G”, where storage modulus related to elastic properties prevails at low angular frequency values (Figure 3d). The complex viscosity was also higher at low angular frequency than in the case of pure starch-containing polylactide (sPLA) (Figure 4d). However, based on the DMA results (Appendix A) starch-containing polylactide with caffeic acid (sPLA/CA) did not change its rheological behavior after UV aging as well as weathering. In the case of starch-containing polylactide (sPLA), caffeic acid unlike green tea extract positively influenced the polymer matrix.

### 3.4. Colour Measurement Results Analysis of Polylactide and Polylctide with Starch Materials before and after Fifferent Types of Aging

Colour is a sensual impression, which depends on different factors, including the quantity and the quality of radiation reflected from the object’s surface, the structure of the eye, mental predisposition, a general sensation or the other colors appearing in the field of view etc. Therefore, UV-Vis spectrophotometry was applied for objective measurement of the material, where light reflected from the object’s surface was analysed. The light source of this apparatus generates light of a wavelength ranging from 360 nm to 740 nm, which is equivalent to the wavelengths that can be sensed by the human eye. The results were obtained in CIE-Lab system, which is an objective colour description. This method is also appropriate for estimating the material colour change as a result of aging process or additive incorporation. The results of colour measurements are presented in Figure 5.

The colour difference corresponds to the distance between two points in the L × a × b colour space. It is believed that the observer sees two different colors, when the colour difference (dE) of compared objects exceeds 5. According to Figure 4, the addition of natural substances to polylactide (PLA) and starch-containing polylactide (sPLA) matrix caused color changing of the neat materials amounting to about 20 except for incorporating caffeic acid to PLA. In that case, the color difference was lower and equaled 5.5. All antioxidants caused an increase in chroma parameter and a decrease in whiteness index. The added natural substances also changed the hue angle of the pure materials, therefore the composites gained new more reddish tones. However, in the case of amorphous PLA, the material did not lose its transparency, as can be noticed in Figure 6c,d which is very important for many packaging producers. Transparent packages are more attractive from the consumer’s point of view, because all the content is visible. It should also be mentioned that in both materials (PLA and sPLA) a good dispersion of green tea extract was achieved but in the case of caffeic acid the aggregates of this antioxidant in polylactide matrix are visible even in the photographs, which indicates unsatisfying additive dispersion. In starch-containing polylactide (sPLA), the dispersion of the same substance was definitely better. Nevertheless, based on Figure 6, green tea extract seems to be a more suitable additive for the tested biopolymers. 

The impact of UV exposure and weathering on colour components was also estimated. Based on the obtained results presented in Figure 7. all materials with natural substances changed their colour as a result of two aging types. Based on Figure 6 and Figure 7, the most significant colour changing occurred in the case of UV exposure. The aged composites with selected natural antioxidants became dramatically darker and gained saturated colors. Based on Figure 6 and Figure 7, the most visible colour modification appeared in polylactide (PLA) and starch-containing polylactide (sPLA) material enriched with green tea extract. Nevertheless, caffeic acid also contributed to appearance modification of the polylactide (PLA) and starch-containing polylactide (sPLA) sample during aging, but only in the case of UV exposure. These observations indicate that colour changing depends on the intensity of irradiation, and humidity does not affect this parameter. It can be noticed that the higher the UV irradiation intensity is, the higher the dE is. The colour changing of green tea extract in starch-containing polylactide and polylactide matrix results in the presence of catechins. These polyphenols have a specific molecular structure characterized by hydroxyl groups at specific positions in the molecule, such as C2–C3 double bond in the C–ring as well as hydroxyl groups at ortho position on the B-ring [43], which are responsible not only for high antioxidant activity but also for UV spectra absorption. Moreover, these substances undergo a photolytic reaction as a result of UV-B radiation exposure, creating colorful products [45]. Therefore, such substances seem to be the candidates for creating UV indicators, which could inform about the exposure of the packaging and its content to the sunlight. Due to the fact that intensive sun radiation is also related to high ambient temperature, these innovative solutions are able to provide information to the consumer about adverse transport or storage conditions. Therefore, such indicators could also help estimate the condition of the packages’ content and make packaging intelligent.

### 3.5. Vicat Softening Temperature of Polylactide (PLA) and Starch-Containing Polylactide (sPLA) Materials before and after Different Types of Aging

The PLA and starch-containing polylactide (sPLA) materials are characterized by the Vicat softening temperature of about 336–337 K, which is consistent with data sheets. The incorporation of natural substances caused a slight decrease in this parameter to 335–336 K in the case of green tea extract (polyphenon 60) addition and to 332–333 K after adding caffeic acid. Biocomposites containing caffeic acid are characterized by the lowest softening temperature which can be also correlated with melting point detected by differential scanning calorimetry (DSC). These composites are able to melt at lower temperatures in comparison to pure bioplastics, and therefore can be better processable. In almost all cases the Vicat softening temperature values raised after the UV exposure as well as weathering except for polylactide enriched with caffeic acid, where this parameter after weathering was equal to the value before aging. According to Figure 8, starch-containing polylactide (sPLA) seems to be more sensitive to UV radiation than PLA, because this parameter was raised dramatically by about 50 K. The impact of weathering on starch-containing polylactide (sPLA) was almost the same, occurring as a slight increase in Vicat softening temperature of about 10 K. 

### 3.6. ATR FT-IR Spectra Analysis of Polylactide (PLA) and Starch-Containing Polylactide (sPLA) Materials Enriched with Substances of Plant Origin before and after UV Exposure and Weathering

The chemical structure of polylactide exposed to UV radiation has recently been analyzed by Stepczynska M. [51] based on the FT-IR spectra obtained. It is assumed that the photodegradation proceeds in the bond between the mers of polylactide. According to Figure 9 and Table 4, the slight modification of PLA and starch-containing polylactide appeared as changes of C–O–C, C=O, C–H bands absorbance as a result of UV radiation and weathering. Moreover, in almost every case, the band of about 801–803 cm^−1^, which is probably related to –C–O–C symmetrical stretching vibration [52] that occurred as a result of the aging process. This band did not appear in the case of PLA with green tea extract, PLA with caffeic acid and starch-containing polylactide with green tea extract after UV exposure, which may signify that these materials are more stable than neat PLA and pure starch-containing polylactide to the action of ultraviolet radiation. In those materials the intensity of C–O–C, C=O, C–H bands after aging processes also did not differ significantly from the absorbance of non-aged composites. Based on the FT-IR spectra, it can be noticed that starch-containing polylactide with caffeic acid was most susceptible to weathering, because the structure changes are most visible. In this material, water attacked the material’s surface, which is observable in the 3367 cm^−1^ absorbance related to the –OH stretching mode and 1633 cm^−1^ of H–O–H bending [53].

## 4. Conclusions

The presented bio-based and biodegradable composites containing natural substances, including an extract from green tea (polyphenon 60) as well as caffeic acid, are able to become intelligent packaging materials. These composites based on commercially available polylactide and starch-containing polylactide (sPLA) have the ability to inform about UV exposure by colour changing. Moreover, depending on the polymer matrix, selected natural additives can act in different ways. Green tea extract can act as a thermal stabilizer in starch-containing polylactide (sPLA) matrix, while in pure PLA it performs better as a UV stabilizer. Caffeic acid also seems to be a sufficient PLA stabilizer, however in starch-containing polylactide (sPLA) matrix the same substance may accelerate the aging of the material in the presence of moisture during weathering. Moreover, caffeic acid is able to change the rheology behavior of the same starch-containing polylactide (sPLA) material. Nevertheless, both of the selected natural substances can be applied as aging indicators for packaging biomaterials to help estimate the condition of the packages’ content.

## Figures and Tables

**Figure 1 materials-13-05719-f001:**
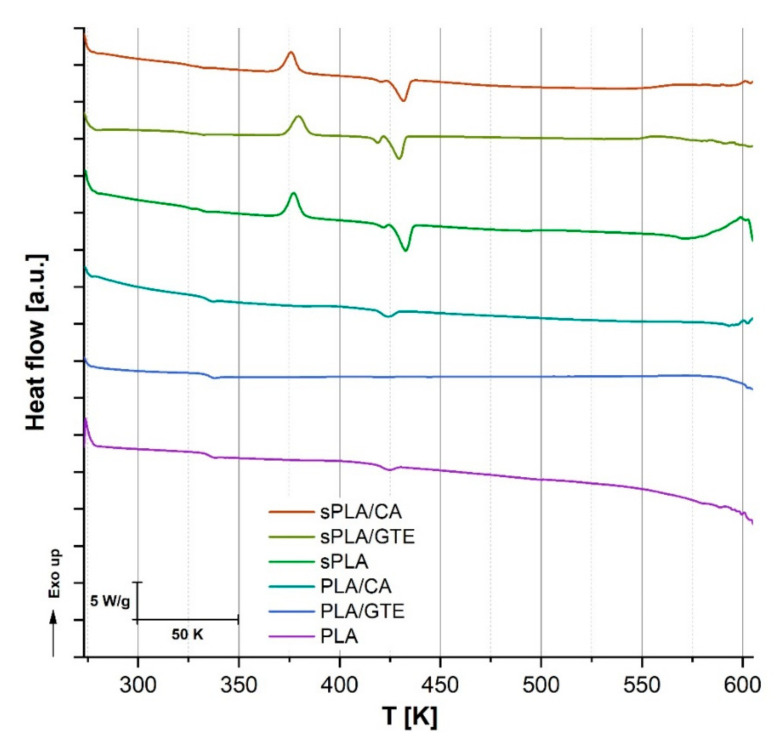
DSC curves of polylactide and polylactide containing starch with selected natural antioxidants in comparison to reference pure PLA and sPLA.

**Figure 2 materials-13-05719-f002:**
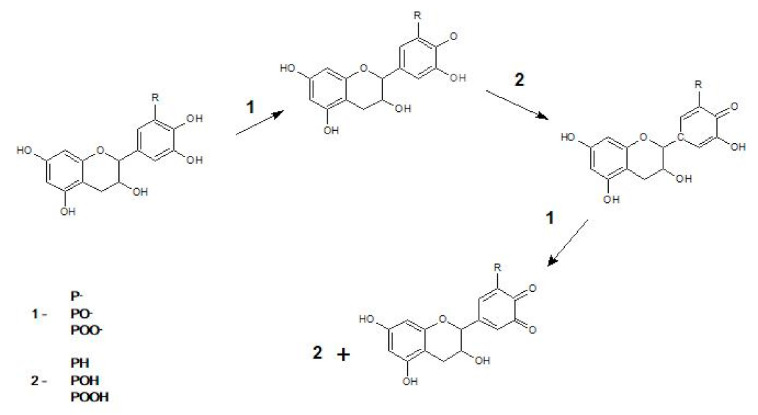
The exemplary mechanism of radical scavenging activity of catechins via hydrogen transfer.

**Figure 3 materials-13-05719-f003:**
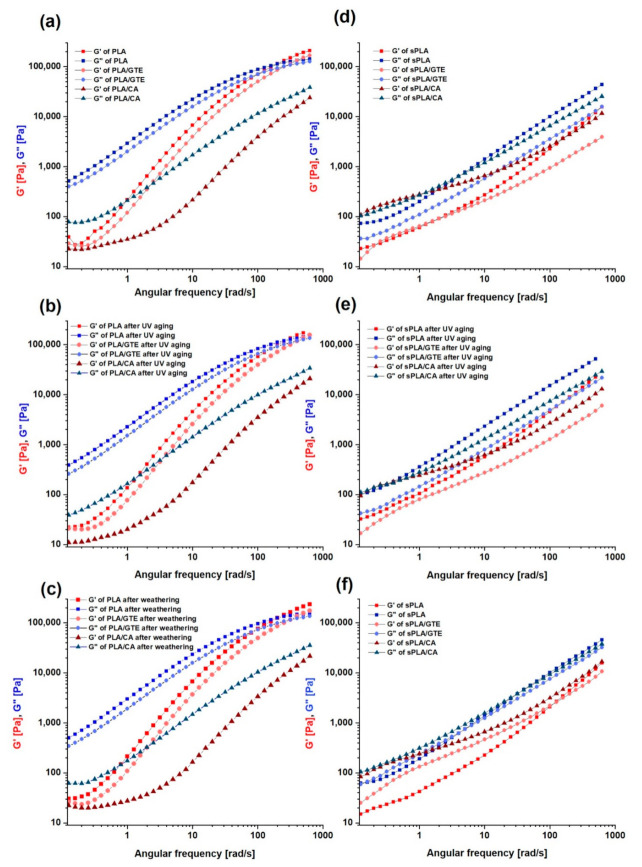
Storage modulus (G’), loss modulus (G”) of: polylactide (PLA), polylactide with green tea extract (PLA/GTE) and polylactide with caffeic acid (PLA/CA) before aging (**a**), after UV exposure (**b**) and after weathering (**c**), as well as starch-containing polylactide (sPLA), polylactide containing starch with green tea extract (sPLA/GTE) and starch-containing polylactide with caffeic acid (sPLA/CA) in a function of angular frequency before (**d**) and after UV aging (**e**) and after weathering (**f**).

**Figure 4 materials-13-05719-f004:**
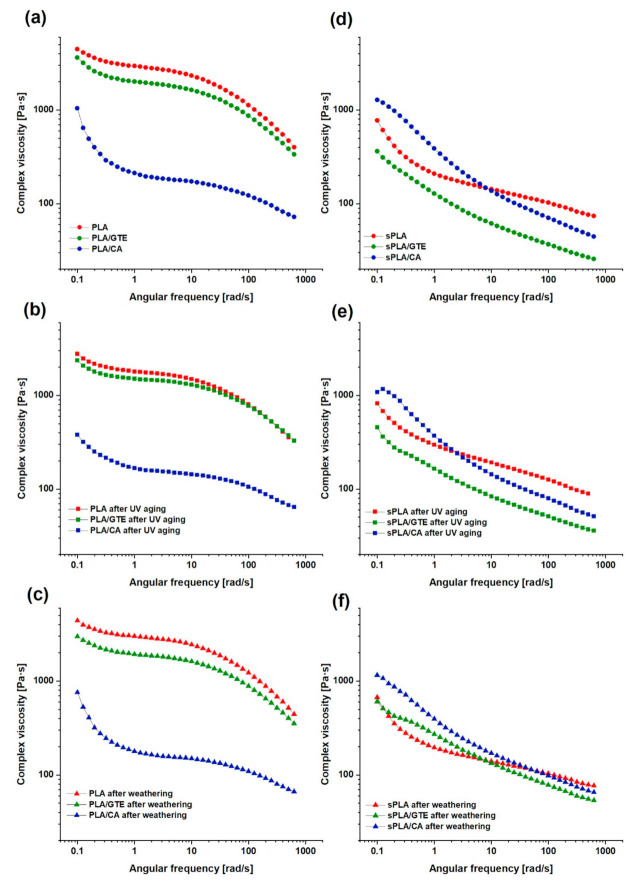
Complex viscosity [Pa × s] of: polylactide, polylactide with green tea extract (PLA/GTE), polylactide with caffeic acid (PLA/CA) before (**a**) and after UV aging (**b**) and after weathering (**c**), starch-containing polylactide (sPLA), polylactide containing starch with green tea extract (sPLA/GTE) and starch-containing polylactide (sPLA)with caffeic acid (sPLA/CA) in a function of angular frequency before (**d**) and after UV aging (**e**) and after weathering (**f**).

**Figure 5 materials-13-05719-f005:**
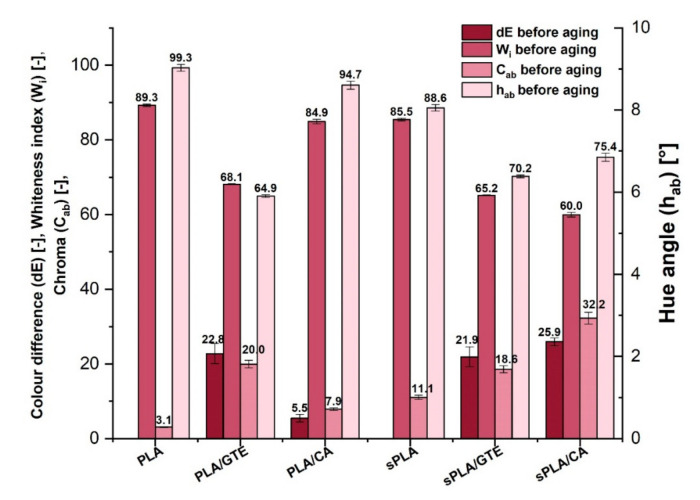
Colour components (W_i_, C_ab_, h_ab_) of polylactide (PLA) and starch-containing polylactide (sPLA) materials and colour difference (dE) after natural substances incorporation.

**Figure 6 materials-13-05719-f006:**
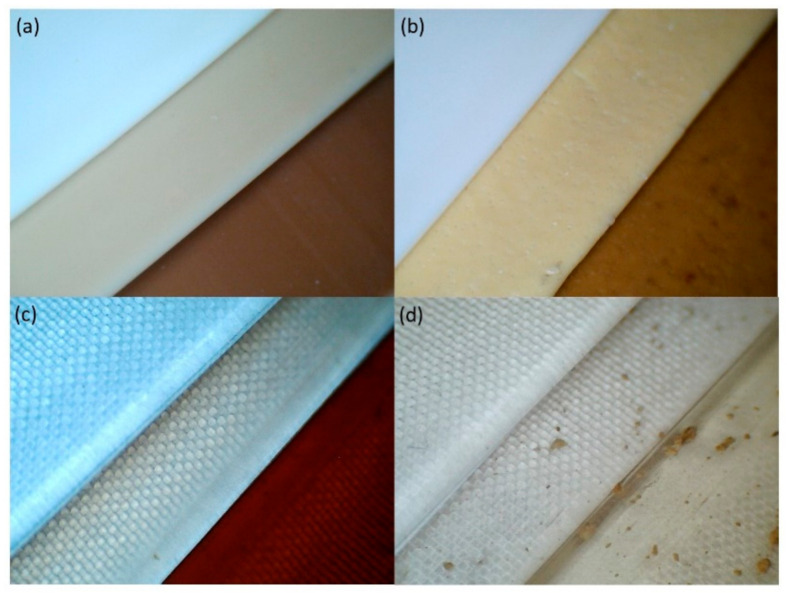
A comparison of pure polylactide (PLA) and starch-containing polylactide (sPLA) materials before and after natural substances incorporations and after UV aging as follows: (**a**) sPLA, starch-containing polylactide with green tea extract (sPLA/GTE), starch-containing polylactide with green tea extract (sPLA/GTE) after UV aging, (**b**) sPLA, starch-containing polylactide with caffeic acid (sPLA/CA), starch-containing polylactide with caffeic acid (sPLA/CA) after UV aging, (**c**) polylactide (PLA), PLA with green tea extract (PLA/GTE), PLA with green tea extract (PLA/GTE) after UV aging, (**d**) polylactide (PLA), polylactide with caffeic acid (PLA/CA), polylactide with caffeic acid (PLA/CA) after UV aging.

**Figure 7 materials-13-05719-f007:**
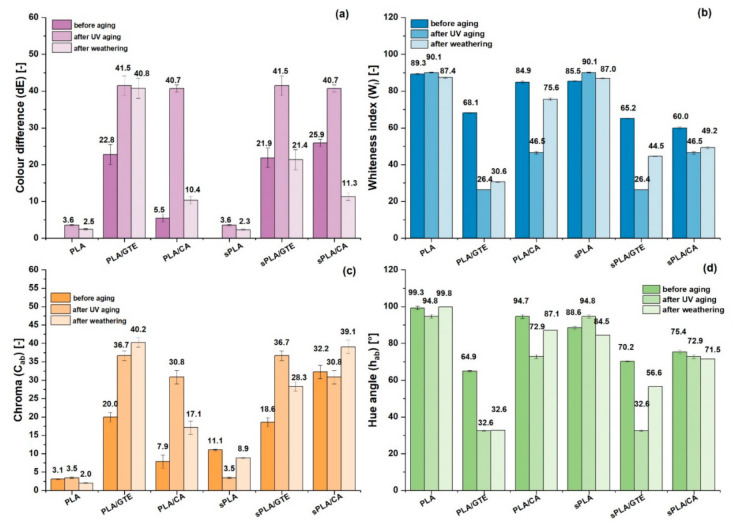
The impact of UV exposure and weathering on: (**a**) change of colour (dE). (**b**) whiteness index (W_i_), (**c**) chroma (C_ab_), (**d**) hue angle (h_ab_) of PLA and starch-containing polylactide (sPLA) materials with an extract from green tea and caffeic acid.

**Figure 8 materials-13-05719-f008:**
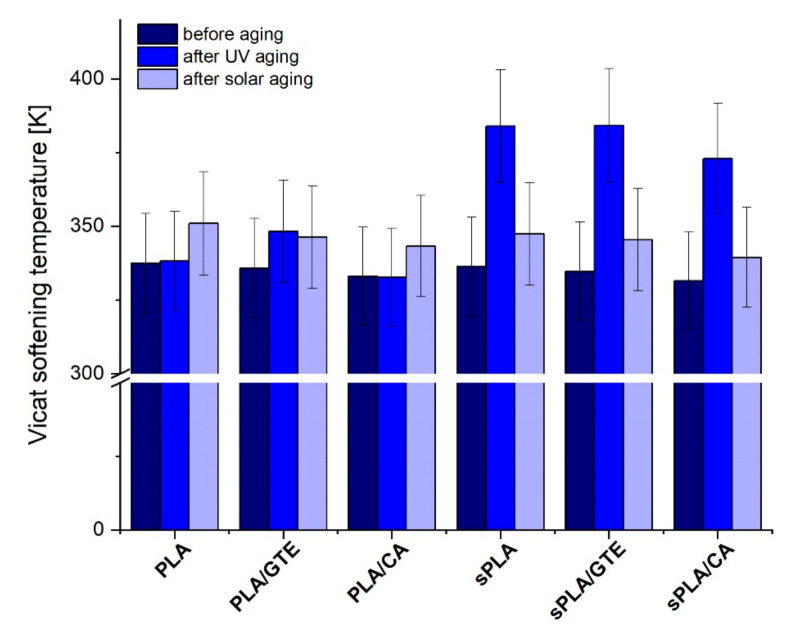
The impact of UV exposure and weathering on Vicat softening temperatures of PLA and starch-containing polylactide (sPLA) materials before and after incorporation of natural antioxidants.

**Figure 9 materials-13-05719-f009:**
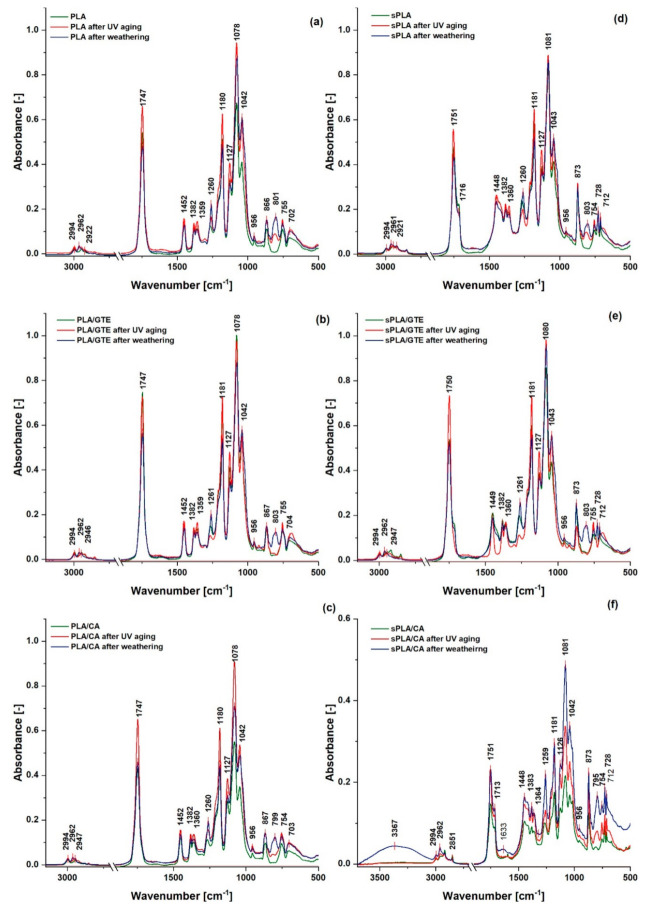
The FT-IR spectra before and after two aging types of: (**a**) polylactide (PLA), (**b**) polylactide with green tea extract (PLA/GTE), (**c**) polylactide with caffeic acid (PLA/CA), (**d**) starch-containing polylactide (sPLA), (**e**) starch-containing polylactide with green tea extract (sPLA/GTE) and (**f**) starch-containing polylactide with caffeic acid (sPLA/CA).

**Table 1 materials-13-05719-t001:** Description of the samples used in the experiment.

Type of Composite	Description
PLA	Polylactide (Ingeo™ Biopolymer 4043D)
PLA/GTE	Polylactide (Ingeo™ Biopolymer 4043D) mixed with green tea extract (polyphenon 60) (mass ratio: 3 parts of antioxidant per 100 parts of polymer pellet)
PLA/CA	Polylactide (Ingeo™ Biopolymer 4043D) mixed with caffeic acid (mass ratio: 3 parts of antioxidant per 100 parts of polymer pellet)
sPLA	Polylactide containing starch (Bioplast GS 2189)
sPLA/GTE	Polylactide containing starch (Bioplast GS 2189) mixed with green tea extract (polyphenon 60) (mass ratio: 3 parts of antioxidant per 100 parts of polymer pellet)
sPLA/CA	Polylactide containing starch (Bioplast GS 2189) mixed with caffeic acid (mass ratio: 3 parts of antioxidant per 100 parts of polymer pellet)

**Table 2 materials-13-05719-t002:** Differential Scanning Calorimetry (DSC) * results of PLA and sPLA enriched with natural antioxidants.

Sample	T_g_ [K]	ΔH_cc_ [J/g]	T_cc_ [K]	ΔH_m_ [J/g]	T_m_ [K]	ΔH_o_ [J/g]	T_o_ [K]
PLA	335	4.7	409	3.1	424	22.6	500
PLA/GTE	335	0.6	408	0.4	424	4.5	546
PLA/CA	334	5.3	404	4.92	423	2.2	552
sPLA	331	16.8	377	19.1	432	17.7	486
sPLA/GTE	333	18.4	380	19.4	430	5.8	549
sPLA/CA	331	16.6	376	18.2	425	1.2	556

* where T_g_—glass transition, ΔH_cc_—cold crystallization enthalpy, T_cc_—temperature of cold crystallization, ΔH_m_—enthalpy of melting, Tm—melting point, ΔH_o_—enthalpy of oxidation process, T_o_—oxidation temperature.

**Table 3 materials-13-05719-t003:** Differential Scanning Calorimetry (DSC) * results of polylactide (PLA) and starch-containing polylactide (sPLA) enriched with natural antioxidants.

Sample	T_5%_ [K]	T_10%_ [K]	T_20%_ [K]	T_40%_ [K]	T_80%_ [K]	T_100%_ [K]
PLA	613	621	630	639	653	867
PLA/GTE	609	618	628	639	653	837
PLA/CA	608	618	628	638	651	780
sPLA	561	574	593	615	769	-
sPLA/GTE	578	590	602	619	751	-
sPLA/CA	551	568	589	612	742	-

* where T_5%_—temperature of 5% material’s weight loss, T_10%_—temperature of 10% material’s weight loss, T_20%_—temperature of 20% material’s weight loss, T_40%_—temperature of 40% material’s weight loss, T_80%_—temperature of 80% material’s weight loss, T_100%_—temperature of 100% material’s weight loss.

**Table 4 materials-13-05719-t004:** Vibrational assignments of polylactide and starch-containing polylactide (sPLA) materials corresponding to the wavenumbers.

Wavenumber [cm^−1^]	Assignments	Ref.
3300	–OH stretching mode (from absorbed water)	[53]
2998–2847	–C–H stretching modes	[54]
1745–1757	–C=O stretching band	[53]
1650	H–O–H bending (from absorbed water)	[53]
1453	–CH_3_ asymmetrical bending	[55]
1360, 1382	–CH symmetrical and asymmetrical bending	[55]
1300	–CH + –C–O–C-	[56]
1268	–C=O bending	[55]
1184, 1130, 1093	–C–O–C asymmetrical streching	[55]
1045	–C–CH_3_ streching	[55]
956	–CH_3_ rocking mode	[56]
868	C–C (amorphous phase)	[54]
801	–C–O–C symmetrical streching vibration-	[52]
756	Crystalline phase	[55]
695	–C=O	[56]

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
