# Peer review of "Bio-Based Packaging Materials Containing Substances Derived from Coffee and Tea Plants"

_materials, 2020, doi:10.3390/ma13245719_

Round 1

Reviewer 1 Report

The manuscript deals with the preparation of biobased composites, making use of natural additives combined with biodegradable materials.

Even if the novelty of the work isn't outstanding, I consider the manuscript suitable for publication after major revisions.

Some comments:

Abstract: authors claim that the composites are more stable when compared to the starting materials, but it is not clear which kind of stability is achieved. Please specify.

Lines 22 to 27 are repeating previous statements. I would suggest to either remove or rephrase them.

Materials and methods: the preparation of the composites is not completely understandable regarding the amount of additive loaded. Not only the targeted loading is not clear, but also the information regarding amount of additive dispersed in the matrix after the extrusion is missing. This last parameter is mandatory in order to fully understand the effect these additives have on the final properties depending on their concentration.

In addition, it would be interesting to have some comments on the uniformity of the samples after the blending process. Did you observe any aggregation phenomena of the additives within the matrix?

Results and discussion: I would suggest to report the DSC thermograms in the manuscript, while the TGA curves are not particularly significant and can be reported only as supporting material.

I really appreciated the data relative to the rheological properties of the tested materials. However, I would suggest to reorganize the curves in order to better highlight the effects of the additives in comparison to the pure materials. For example, I would regroup the curves to compare the pure materials and the blends in the same conditions, like:

a) Moduli of PLA, PLA/polyphenon 60 and PLA/caffeic

b) Moduli of PLA, PLA/polyphenon 60 and PLA/caffeic all after UV aging

c) Moduli of PLA, PLA/polyphenon 60 and PLA/caffeic all after UV weathering

and so on.

Author Response

Institute of Polymer and Dye Technology

Technical University of Lodz

90-924 Lodz, ul Stefanowskiego 12/16, Poland

Tel.: +48 42 631 32 23, Fax: +48 42 636 25 43

November 26, 2020

Polymers

Dear Professor,

We are resubmitting our revised paper entitled Bio-based packaging materials containing substances derived from coffee and tea plants by, Olga Olejnik, Anna Masek with a request to reconsider it for publication in Materials.

We have carefully considered the Editor and Reviewers' comments. The manuscript was revised exactly according to these comments. The list of responses to the reviewers’ comments and corrections made in the manuscript is attached.

The manuscript has not been previously published, is not currently submitted for review to any other journal, and will not be submitted elsewhere before a decision is made by this journal.

For correspondence please use the following information:

corresponding author: Anna Masek

Institute of Polymer and Dye Technology

Technical University of Lodz

90-924 Lodz, ul Stefanowskiego 12/16, Poland

Tel.: +48 42 631 32 93

Fax: +48 42 636 25 43

e-mail: anna.masek@p.lodz.pl

Yours sincerely,

Ph. D., D.Sc. Anna Masek

Answers to reviewer #1 comments

Reviewer #1: The manuscript deals with the preparation of biobased composites, making use of natural additives combined with biodegradable materials. Even if the novelty of the work isn't outstanding, I consider the manuscript suitable for publication after major revisions. Some comments:

  1. Abstract:authors claim that the composites are more stable when compared to the starting materials, but it is not clear which kind of stability is achieved. Please specify.

Lines 22 to 27 are repeating previous statements. I would suggest to either remove or rephrase them.

Answer 1.: We are thankful for the Reviewer’s comment. The abstract has already been improved. The achieved kind of stability is highlighted and repeated information has been removed. Here, we present the new abstract:

The aim of the research was to obtain intelligent and eco-friendly packaging materials by incorporating innovative additives of plant origin. For this purpose, natural substances, including green tea extract (polyphenon 60) and caffeic acid were added to two types of biodegradable thermoplastics (Ingeo™ Biopolymer PLA 4043D and Bioplast GS 2189). The main techniques used to assess the impact of phytocompounds on materials’ thermal properties were Differential Scanning Calorimetry (DSC) and Thermogravimetry (TGA), which confirmed the improved resistance to thermo-oxidation. Moreover, in order to assess the activity of applied antioxidants, the samples were aged using a UV aging chamber and a weathering device, then retested in terms of Dynamic Mechanical properties (DMA), colour changing, Vicat softening temperature and chemical structure studied using FT-IR spectra analysis. The results revealed that different types of aging did not cause significant differences in thermo-mechanical properties and chemical structure of the samples with natural antioxidants but induced colour changing. The obtained results indicate that polylactide (PLA) and Bioplast GS 2189, the plasticizer free thermoplastic biomaterial containing polylactide and starch (referred to as sPLA in the present article), both with added caffeic acid and green tea extract, can be applied as smart and eco-friendly packaging materials. The composites reveal better thermo-oxidative stability with reference to pure materials and are able to change the colour as a result of oxidation process, especially after UV exposure, informing about the degree of material degradation.

  1. Materials and methods:the preparation of the composites is not completely understandable regarding the amount of additive loaded. Not only the targeted loading is not clear, but also the information regarding amount of additive dispersed in the matrix after the extrusion is missing. This last parameter is mandatory in order to fully understand the effect these additives have on the final properties depending on their concentration. In addition, it would be interesting to have some comments on the uniformity of the samples after the blending process. Did you observe any aggregation phenomena of the additives within the matrix?

Answer 2.:We thank the Reviewer for paying attention to these problems. We have added Table 1., which presents a clear description of composites preparation, including the amount of additive. Moreover, we added information about dispersion based on the photographs of materials in chapter “3.4. Colour measurement results analysis of polylactide and polylctide with starch materials before and after different types of aging” as follows:

It should also be mentioned that in both materials (PLA and sPLA) a good dispersion of green tea extract was achieved but in the case of caffeic acid the aggregates of this antioxidant in polylactide matrix are visible even in the photographs, which indicates unsatisfying additive dispersion. In polylactide containing starch (sPLA), the dispersion of the same substance was definitely better. Nevertheless, based on Figure 5. green tea extract seems to be a more suitable additive for the tested biopolymers.

  1. Results and discussion:I would suggest to report the DSC thermograms in the manuscript, while the TGA curves are not particularly significant and can be reported only as supporting material.

Answer 3.: We are thankful for the Reviewer’s advice. We moved TGA curves to the Supplementary Material and in the article we added DSC thermograms instead.

Figure 1. DSC curves of polylactide and polylactide containing starch with selected natural antioxidants in comparison to reference pure PLA and sPLA.

  1. I really appreciated the data relative to the rheological properties of the tested materials. However, I would suggest to reorganize the curves in order to better highlight the effects of the additives in comparison to the pure materials. For example, I would regroup the curves to compare the pure materials and the blends in the same conditions, like:
  2. a) Moduli of PLA, PLA/polyphenon 60 and PLA/caffeic
  3. b) Moduli of PLA, PLA/polyphenon 60 and PLA/caffeic all after UV aging
  4. c) Moduli of PLA, PLA/polyphenon 60 and PLA/caffeic all after UV weathering

and so on.

Answer 4.: We appreciate Reviewer’s suggestions. We changed the curves in order to better highlight the effects of the additives in comparison to the pure materials. Nevertheless, we added the previous version of these figures to Supplementary Materials to better underline effects of the different aging processes on the prepared composites.

Figure 2. Storage modulus (G’), loss modulus (G”) of: polylactide (PLA), polylactide with green tea extract (PLA/GTE) and polylactide with caffeic acid (PLA/CA) before aging (a), after UV exposure (b) and after weathering (c), as well as polylacide containing starch (sPLA), polylactide containing starch with green tea extract (sPLA/GTE) and polylactide containing starch with caffeic acid (sPLA/CA) in a function of angular frequency before (d) and after UV aging (e) and after weathering (f).

Figure 3. Complex viscosity [Pa·s] of: polylactide, polylactide with green tea extract (PLA/GTE), polylactide with caffeic acid (PLA/CA), polylactide containing starch (sPLA), polylactide containing starch with green tea extract (sPLA/GTE) and polylactide containing starch with caffeic acid (sPLA/CA) in a function of angular frequency before (d) and after UV aging (e) and after weathering (f).

Reviewer 2 Report

Review comments:

1) English usage should be checked. for example - Page 3 line 107, "The producer ensures that... " is not technical English in a scientific paper.

2) Why did authors only use only one temperature for different zones during extrusion (Page 3 line 122)? 

3) What are actual strength and modulus properties of the composites from bending, tensile and/or impact testing? These properties are more relevant to reflect composite properties.

4) What is the micro-morphology of the composites? how was starch distributed in the composites?

5) some mechanism discussion is needed on the effect of selected antioxidants of plant origin. 

6) The claim of intelligent packaging for the developed materials needs to be further justified. 

Author Response

Institute of Polymer and Dye Technology

Technical University of Lodz

90-924 Lodz, ul Stefanowskiego 12/16, Poland

Tel.: +48 42 631 32 23, Fax: +48 42 636 25 43

November 26, 2020

Polymers

Dear Professor,

We are resubmitting our revised paper entitled Bio-based packaging materials containing substances derived from coffee and tea plants by, Olga Olejnik, Anna Masek with a request to reconsider it for publication in Materials.

We have carefully considered the Editor and Reviewers' comments. The manuscript was revised exactly according to these comments. The list of responses to the reviewers’ comments and corrections made in the manuscript is attached.

The manuscript has not been previously published, is not currently submitted for review to any other journal, and will not be submitted elsewhere before a decision is made by this journal.

For correspondence please use the following information:

corresponding author: Anna Masek

Institute of Polymer and Dye Technology

Technical University of Lodz

90-924 Lodz, ul Stefanowskiego 12/16, Poland

Tel.: +48 42 631 32 93

Fax: +48 42 636 25 43

e-mail: anna.masek@p.lodz.pl

Yours sincerely,

Ph. D., D.Sc. Anna Masek

Answers to reviewer #2 comments

Reviewer #2:1) English usage should be checked. for example - Page 3 line 107, "The producer ensures that... " is not technical English in a scientific paper.

Answer 1.: We are thankful for this comment. We tried to improve technical English in the paper. For example we changed "The producer ensures that... " into “In the technical data sheet of this polymer it can be found that this material in a granulate form is easyflowing…”

2) Why did authors only use only one temperature for different zones during extrusion (Page 3 line 122)?

Answer 2.: We thank the Reviewer for paying attention to this issue. In our laboratory extruder the different temperatures in all zones cannot be kept. These zones are too close and heat up each other, thus we obtain the same temperature in every zone. Nevertheless, the presented parameters provided the best processability of our composites.

3) What are actual strength and modulus properties of the composites from bending, tensile and/or impact testing? These properties are more relevant to reflect composite properties.

Answer 3.: We are thankful for the Reviewer’s comment. At that moment we were focused more on the thermal properties of these composites than mechanical ones. Moreover, the proposed tests require also a larger amount of material than we could prepare. Also the type of composites preparing must be changed to obtain the required specimens for this test, thus the results would not be compatible with results from current research.

4) What is the micro-morphology of the composites? how was starch distributed in the composites?

Answer 4.: We are grateful for paying attention to this problem by the Reviewer. Unfortunately we do not have SEM images to show the micro-morphology of the composites. At the moment the micro-morphology description of composites is not possible.

5) some mechanism discussion is needed on the effect of selected antioxidants of plant origin.

Answer 5.: We appreciate the Reviewer’s suggestions. We added a mechanism explaining the stabilizing effect of selected natural antioxidants exemplifying it by catechins present in the extract of green tea (polyphenon 60) as follow:

Figure 1. The exemplary mechanism of radical scavenging activity of catechins via hydrogen transfer.

The delayed oxidation process detected by Differential Scanning Calorimetry can be explained by mechanism of radical scavenging activity via hydrogen transfer, presented in Figure 1. This mechanism is characteristic of catechins and other polyphenols present in green tea extract applied in this research. The caffeic acid also is a natural phenolic compound with antioxidant properties, thus also reveals a similar stabilizing effect.

6) The claim of intelligent packaging for the developed materials needs to be further justified.

Answer 6.: We appreciate Reviewer’s suggestions. We decided to add some information about the potential of our composites as intelligent food packaging materials. The information was added to the introduction:

“Moreover, catechins and caffeic acid present in tea and coffee extracts change their colour after UV exposure [46], thus these natural phenolic compounds after incorporation to the polymer could inform about the exposure of the packaging and its content to the sunlight. Therefore combination of polymeric material with selected substances of plant origin can result in the creation of intelligent food packaging.”

Reviewer 3 Report

Exemplification through studies and results is a factor that gives credibility to the study under investigation, highlights the benefits of implementing such a system, but does not specify or mention the openness of the entities involved to implement systems that can lead to these results. I would thus suggest an additional addition (even only descriptive-a few lines) through which the authors highlight the real possibilities of implementing their results, from the perspective of the actors involved.

Author Response

Institute of Polymer and Dye Technology

Technical University of Lodz

90-924 Lodz, ul Stefanowskiego 12/16, Poland

Tel.: +48 42 631 32 23, Fax: +48 42 636 25 43

November 26, 2020

Polymers

Dear Professor,

We are resubmitting our revised paper entitled Bio-based packaging materials containing substances derived from coffee and tea plants by, Olga Olejnik, Anna Masek with a request to reconsider it for publication in Materials.

We have carefully considered the Editor and Reviewers' comments. The manuscript was revised exactly according to these comments. The list of responses to the reviewers’ comments and corrections made in the manuscript is attached.

The manuscript has not been previously published, is not currently submitted for review to any other journal, and will not be submitted elsewhere before a decision is made by this journal.

For correspondence please use the following information:

corresponding author: Anna Masek

Institute of Polymer and Dye Technology

Technical University of Lodz

90-924 Lodz, ul Stefanowskiego 12/16, Poland

Tel.: +48 42 631 32 93

Fax: +48 42 636 25 43

e-mail: anna.masek@p.lodz.pl

Yours sincerely,

Ph. D., D.Sc. Anna Masek

Answers to reviewer #3 comments

Reviewer #3: Exemplification through studies and results is a factor that gives credibility to the study under investigation, highlights the benefits of implementing such a system, but does not specify or mention the openness of the entities involved to implement systems that can lead to these results. I would thus suggest an additional addition (even only descriptive-a few lines) through which the authors highlight the real possibilities of implementing their results, from the perspective of the actors involved.

Answer 1.: We are thankful for this comment. We added some information about application composites in food packaging sector, as follows:

“Especially the food packaging sector is showing an increasing interest in the development of innovative solutions, including new processes and materials. The natural substances, including phenols from plant extracts may chance to play a significant role in active as well as intelligent food packaging. Nevertheless most of proposed natural additives are derived from fruit or vegetables [43]. Here we present the potential of polyphenolic extract derived from green tea, as well as a phenolic acid present in coffee plant, as multifunctional auxiliary substances dedicated to pro-ecological packaging materials. These substances have good antioxidant properties [48,49], thus seem to be good active ingredients for polymers, especially biodegradable ones dedicated for food packaging industry. Moreover, catechins and caffeic acid present in tea and coffee extracts change their colour after UV exposure [46], thus these natural phenolic compounds after incorporation to the polymer could inform about the exposure of the packaging and its content to the sunlight. Therefore a combination of polymeric material with selected substances of plant origin can result in the creation of intelligent food packaging.”

Round 2

Reviewer 1 Report

I wish to thank the authors for addressing my concerns.

I now believe that the manuscript can be accepted for publication.

Reviewer 2 Report

can be published as it is now.